# Classical Mechanics with Inequality Constraints and Gravity Models with Limiting Curvature

**Andrei V. Frolov** [1,†] and **Valeri P. Frolov** [2,*,†]

1  Department of Physics, Simon Fraser University, Burnaby, BC V5A 1S6, Canada; frolov@sfu.ca
2  Theoretical Physics Institute, University of Alberta, Edmonton, AB T6G 2E1, Canada
*  Correspondence: vfrolov@ualberta.ca
†  These authors contributed equally to this work.

**Abstract:** In this paper, we discuss mechanical systems with inequality constraints $\Phi(q, \dot{q}, ...) \leq 0$. We demonstrate how such constraints can be taken into account by proper modification of the action which describes the original unconstrained dynamics. To illustrate this approach, we consider a harmonic oscillator in the model with limiting velocity. We compare the behavior of such an oscillator with the behavior of a relativistic oscillator and demonstrate that when the amplitude of the oscillator is large, the properties of both types of oscillators are quite similar. We also discuss inequality constraints, which contain higher derivatives. At the end of the paper, we briefly discuss possible applications of the developed approach to gravity models with limiting curvature.

**Keywords:** classical mechanics; Lagrangian formulation; limiting curvature; gravity models

## 1. Introduction

Studying mechanical systems with constraints has a long history (see for example [1,2]). For such systems besides the Lagrangian $L(q, \dot{q})$, there exist one or more constraint relationships $\Phi_i = 0$ which restrict a motion of the system. In the simplest case, when these functions depend only on the coordinates $q$, the constraints are called holonomic [1]. By solving these constraints, one can reduce the configuration space and obtain a reduced Lagrangian which describes the motion of the system in the presence of the constraints. A case of nonholonomic constraints, where constraint function $\Phi_i$ depends not only on the coordinates $q$ but also on velocities $\dot{q}$, is more complicated. There exist many publications devoted to this subject. The discussion of this problem and corresponding references can be found, for example, in [3–8].

One of the methods to study mechanical systems with nonholonomic constraints was developed in [9]. In this approach, one "upgrades" the Lagrangian by adding to it the constraints with corresponding Lagrange multipliers. This method, which uses a generalization of Hamilton's principle of stationary action, was coined as *vakonomic mechanics* [6–10]. For a system with one constraint $\Phi(q, \dot{q}) = 0$, the corresponding "upgraded" Lagrangian is

$$\mathcal{L} = L(q, \dot{q}) + \chi \Phi(q, \dot{q}) \,. \tag{1}$$

The variation of the action with respect to Lagrange multiplier $\chi$ reproduces the constraint equation $\Phi = 0$, while its variation over $q$ gives equations containing, besides the coordinates, the control function $\chi(t)$. This set of equations determines the constrained motion of the system.

The purpose of this paper is to discuss mechanics with inequality constraints. Such a constraint is a restriction imposed on coordinates $q$ and velocities $\dot{q}$, which are of the form $\Phi(q, \dot{q}) \leq 0$. We shall demonstrate that the equations describing the system's motion can be obtained from the following Lagrangian

$$\mathcal{L} = L + \chi(\Phi + \zeta^2) \,, \tag{2}$$

which contains two Lagrange multipliers, $\chi$ and $\zeta$. Variation of the corresponding action over $\chi$ and $\zeta$ gives

$$\Phi + \zeta^2 = 0, \quad \chi\zeta = 0. \tag{3}$$

As a result, the motion of the system has two different regimes. If $\Phi \leq 0$, the first of these equations defines $\zeta$, while the second equation implies $\chi = 0$. When the first equation is saturated and the constraint function $\Phi$ reaches its critical value, $\Phi = 0$, one has $\zeta = 0$, and the control function $\chi$ can take a non-zero value. A transition between these two regimes occurs at points where the control function $\chi$ vanishes. A similar approach was discussed in [11]. In recent publications [12–15], models with inequality constraints imposed on the curvature invariants and their application to the problem of black hole and cosmological singularities were discussed.

The paper is organized as follows. In Section 2, we consider the mechanics of a system with one degree of freedom and with an inequality constraint. In Section 3, we discuss transitions between different regimes and find the corresponding conditions. In Section 4, we apply the developed tools to study a harmonic oscillator with an inequality restriction imposed on its velocity. In Section 5, we compare the motion of the oscillator in the limiting velocity model with a motion of a relativistic oscillator, and demonstrate that there exists a similarity between these two models. A case when the constraint function $\Phi$ contains higher than first derivatives of coordinates is briefly discussed in Section 6. The last section summarizes the obtained results and discusses their applications to the recently proposed models of gravity with the limiting curvature.

## 2. Lagrangian Mechanics with an Inequality Constraint

Our starting point is the following action

$$S = \int dt\, \mathcal{L}, \tag{4}$$

where

$$\mathcal{L}(q, \dot{q}, \chi, \zeta) = L(q, \dot{q}) + \chi\left(\Phi(q, \dot{q}) + \zeta^2\right). \tag{5}$$

Here and later, we denote by dot the time derivative. This action, besides the dynamical variable $q(t)$, contains a constraint function $\Phi(q, \dot{q})$ and two Lagrange multipliers $\chi(t)$ and $\zeta(t)$. The variation of $S$ with respect to $\chi(t)$ and $\zeta(t)$ gives

$$\Phi(q, \dot{q}) + \zeta^2 = 0, \quad \chi\zeta = 0. \tag{6}$$

These equations imply that the evolution of the system can have two different regimes, namely:

- Subcritical phase, where the constraint function $\Phi(q, \dot{q})$ is negative. In this regime, the first equation in (6) defines the Lagrange multiplier $\zeta(t)$, while the second relationship implies that $\chi(t) = 0$. This means that during the subcritical phase, the system follows the standard Euler–Lagrange equation.
- Supercritical regime, where $\zeta = 0$. At this phase, the constraint equation

$$\Phi(q, \dot{q}) = 0 \tag{7}$$

  is satisfied and the control parameter $\chi(t)$ becomes dynamical.

If the system described by action (4) starts its motion in the subcritical regime, then its motion is described by the Euler–Lagrange equation

$$\frac{\delta L}{\delta q} = 0. \tag{8}$$

Here and later, we use the following standard definition of a Lagrange derivative of a function $L(q, \dot{q})$

$$\frac{\delta L}{\delta q} \equiv \frac{d}{dt}\left(\frac{\partial L}{\partial \dot{q}}\right) - \frac{\partial L}{\partial q}. \tag{9}$$

Let $q(t)$ be a solution of this equation, then the constraint function $\Phi(t) = \Phi(q(t), \dot{q}(t))$ calculated on the trajectory is negative. If at some time $t_0$ the constraint function $\Phi(t)$ vanishes, the supercritical regimes starts.

At this stage, the constraint $\Phi = 0$ is valid and the variation of action (4) over $q$ gives

$$\dot{\chi}\pi + \chi\frac{\delta\Phi}{\delta q} + \frac{\delta L}{\delta q} = 0. \tag{10}$$

Here we denote

$$\pi \equiv \frac{\partial\Phi}{\partial\dot{q}}, \qquad \frac{\delta\Phi}{\delta q} \equiv \frac{d}{dt}\left(\frac{\partial\Phi}{\partial\dot{q}}\right) - \frac{\partial\Phi}{\partial q}. \tag{11}$$

A motion in this phase is specified by two functions of time, $q(t)$ and $\chi(t)$. Function $q(t)$ can be found by solving the constraint Equation (7), while Equation (10) determines the evolution of the control function $\chi(t)$. This is a first-order ordinary differential equation for $\chi(t)$ and its solution is determined if one specifies a value of the control function $\chi$ at some moment of time.

If the constraint function $\Phi$ reaches its critical value 0 at some time $t = t_0$, then $\chi(t_0) = 0$. If we assume that $\pi(t_0) \neq 0$, then Equation (10), with this initial condition, has a unique solution. The system can return to its subcritical regime if the control function $\chi(t)$ vanishes again at some later time $t_1 > t_0$.

## 3. Condition of the Regime Change

Let us discuss a condition when the supercritical solution can return to the subcritical one in more detail. For this purpose, we introduce the following notation:

$$p = \frac{\partial L}{\partial\dot{q}}, \quad E = p\dot{q} - L, \quad \epsilon = \pi\dot{q} - \Phi. \tag{12}$$

Let us demonstrate that in the supercritical regime the quantity

$$\mathcal{E} = E + \chi\epsilon, \tag{13}$$

is conserved.

It is easy to check that the following relationships are valid:

$$\dot{E} = \frac{\delta L}{\delta q}\dot{q}, \qquad \dot{\epsilon} = \frac{\delta\Phi}{\delta q}\dot{q}. \tag{14}$$

Using the definition of $\mathcal{E}$ and relationships (14), one has

$$\dot{\mathcal{E}} = \dot{E} + \chi\dot{\epsilon} + \dot{\chi}\epsilon = \left(\frac{\delta L}{\delta q} + \chi\frac{\delta\Phi}{\delta q} + \dot{\chi}\pi\right)\dot{q} - \dot{\chi}\Phi. \tag{15}$$

Using relationship (10) and the constraint equation $\Phi = 0$, one obtains $\dot{\mathcal{E}} = 0$.

The obtained equality means that $\mathcal{E}$ is a conserved quantity for the supercritical solution. By its construction, it has the meaning of the energy of the enlarged system described by the extended action (4) and (5). It differs from the energy $E$ of the unconstrained system by the quantity $\chi\epsilon$. The latter can be interpreted as the contribution of the control field $\chi$ to the total energy of the system. In the subcritical regime, $\chi = 0$ and $E = E_0 = \text{const}$, where $E$ is the usual energy of the unconstrained system. At the transition point $t = t_0$, where $\chi = 0$, one has $\mathcal{E} = E_0$.

During the supercritical phase, the quantity $E$ depends on time, $E = E(t)$. It is easy to see that, if there exists a later moment of time $t = t_1$ where the control parameter $\chi$ becomes zero again, the following condition should be valid

$$E(t_1) = E_0 . \tag{16}$$

In other words, the supercritical solution can return to the subcritical regime at the point where the total energy $\mathcal{E}$ becomes equal to the energy $E_0$ of the system in the initial subcritical phase.

Figure 1 illustrates a motion of a system with the inequality constraint. It shows curves of constant energy $E$ and of the constraint $\Phi = 0$ on the $(q, \dot{q})$ phase plane. Let $ABC$ be $E = \text{const}$ curve ($E$-curve) and $DEF$ be a constraint curve $\Phi = 0$. Point 0 and 1 where these curves intersect are transition points between the sub- and supercritical regimes. Suppose that $\Phi$ is positive above the curve $DEF$. Then, the system moving from $A$ after reaching the transition point 0 moves along supercritical trajectory $0E1$. After the second transition point 1 it continues its motion along a subcritical trajectory $1C$. Since the velocity $\dot{q}$ on $0E1$ is less than the velocity on $0B1$ for the same value of $q$, the time of "travel" between 0 and 1

$$t_{01} = \int_{q_0}^{q_1} \frac{dq}{\dot{q}} \tag{17}$$

along the constraint curve is longer than the corresponding time of motion along the $E$-curve. In the opposite case, where $\Phi$ is positive below the curve $DEF$, the corresponding trajectory is $D0B1F$.

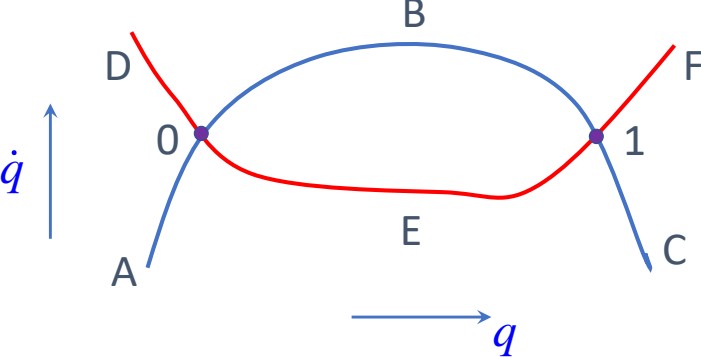

**Figure 1.** Curves of constant energy, $E = \text{const}$, and constraint $\Phi = 0$ on the $(q, \dot{q})$ phase plane.

## 4. Harmonic Oscillator with the Limiting Velocity Constraint

*4.1. Motion of Oscillator with the Velocity Constraint*

To illustrate the results described in the previous section, let us consider a model of a harmonic oscillator with the limiting velocity constraint. Specifically, we discuss a system with the following action:

$$\tilde{S} = \frac{1}{2} \int d\tau \left[ (dx/d\tau)^2 - \omega^2 x^2 + \chi \left( (dx/d\tau)^2 - V^2 + \tilde{\zeta}^2 \right) \right] . \tag{18}$$

Here, $x(\tau)$ is a position of the oscillator at time $\tau$ and $\omega$ is its frequency. The constant $V$ is the value of the limiting velocity and a constraint is chosen so that the following inequality: $|dx/d\tau| \leq V$ is valid. It is convenient to introduce dimensionless variables

$$q = \omega x / V, \quad t = \omega \tau, \quad \zeta = \tilde{\zeta} / V \tag{19}$$

in which

$$\tilde{S} = \frac{V^2}{\omega} S, \quad S = \frac{1}{2} \int dt \left[ \dot{q}^2 - q^2 + \chi (\dot{q}^2 - 1 + \zeta^2) \right] . \tag{20}$$

Here, the dot means a derivative with respect to the dimensionless time $t$.

The variation of the action $S$ over $q$, $\chi$ and $\zeta$ gives the following set of equations:

$$(\dot{q}\chi)^{\cdot} = -(\ddot{q} + q), \quad \dot{q}^2 - 1 + \zeta^2 = 0, \quad \zeta\chi = 0. \tag{21}$$

In the subcritical regime, $\chi = 0$ and $\zeta^2 = 1 - \dot{q}^2$, while Equation (21) reproduces the standard equation of motion of a non-relativistic harmonic oscillator

$$\ddot{q} + q = 0. \tag{22}$$

In the supercritical regime

$$\dot{q}^2 = 1, \quad \dot{q}\dot{\chi} + \ddot{q}\chi = -(\ddot{q} + q). \tag{23}$$

At the transition point separating these regimes, $|\dot{q}| = 1$ and $\chi = 0$.

We choose a solution in the subcritical regime in the form

$$q = -q_0 \cos(t), \quad q_0 > 0. \tag{24}$$

Then, a transition to the supercritical regime occurs when

$$\dot{q} \equiv q_0 \sin(t) = \pm 1. \tag{25}$$

This relationship can be satisfied only if $q_0 \geq 1$. This means that, for $q_0 < 1$, a solution always remains in the subcritical regime and the limiting velocity is not reached, so the motion is that of a simple harmonic oscillator.

For $q_0 > 1$, solution (24) reaches the critical velocity $\dot{q} = 1$ for the first time at the moment $t = t_0$

$$t_0 = \arcsin(1/q_0). \tag{26}$$

A position of the oscillator at this moment is

$$q(t_0) = -q_c, \quad q_c = \sqrt{q_0^2 - 1}. \tag{27}$$

After this time, the oscillator follows the supercritical trajectory

$$q(t) = -q_c + (t - t_0). \tag{28}$$

Equation (23), which determines the evolution of the control function, takes the form

$$\dot{\chi} = -q. \tag{29}$$

Using (28) and solving Equation (29) with the initial condition $\chi(t_0) = 0$, one finds

$$\chi = -\frac{1}{2}(t - t_0)^2 + q_c(t - t_0). \tag{30}$$

The control function vanishes again at time $t = t_1 > t_0$, where

$$t_1 = t_0 + 2q_c. \tag{31}$$

After this, the solution returns to its subcritical regime where $\chi = 0$ and

$$q = q_0 \cos[2(q_c + t_0) - t]. \tag{32}$$

This subcritical motion continues until the moment of time $t = t_2 > t_1$, when the velocity reaches the critical value $\dot{q} = -1$. For $t_2 < t < t_3$, the oscillator moves with a constant critical velocity $\dot{q} = -1$ until a new transition to the subcritical regime occurs at $t = t_3$.

The phase diagram for the harmonic oscillator in the limiting velocity model is shown in Figure 2. The inner circle $C_<$ describes the motion of the oscillator with $q_0 < 1$. For this case, the supercritical regime is absent. The outer orbit $C_>$ describes the motion of the oscillator with $q_0 > 1$. Points 0, 1, 2, and 3 represent the states where the transitions between subcritical and supercritical phases occur.

Figure 3 shows the plot of the function $q(t)$ for the oscillator with the velocity constraint (a solid blue line) and for the standard harmonic oscillator with the same amplitude $q_0 > 1$ (a dashed red line).

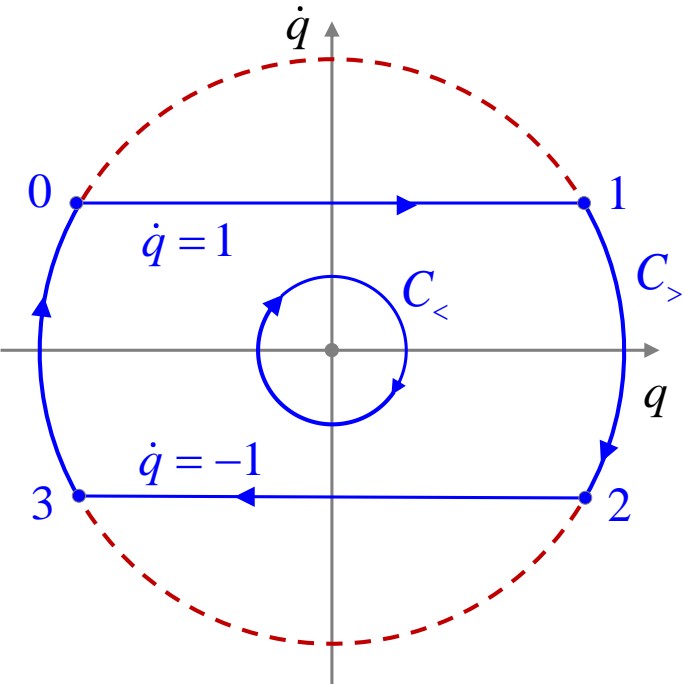

**Figure 2.** Phase diagram on $(q, \dot{q})$-plane representing the motion of an oscillator in the limiting velocity model. These plots are shown for two values of the amplitude $q_0 = 0.5$ (line $C_<$) and $q_0 = 2.0$ (line $C_>$).

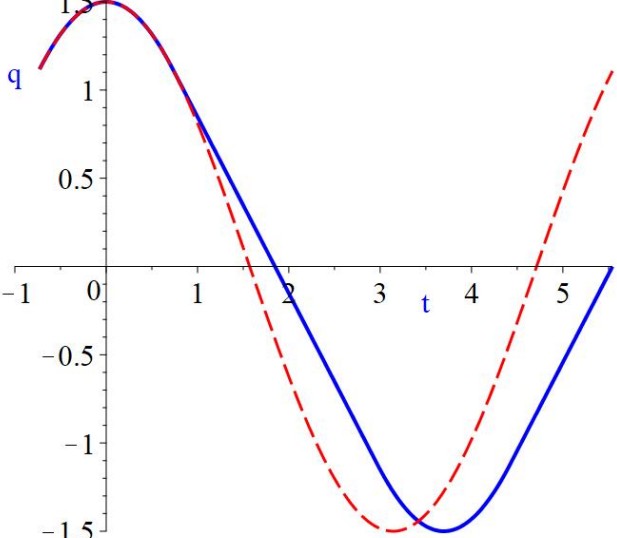

**Figure 3.** Time dependence of a position of the oscillator with amplitude $q_0 = 1.5$ in the limiting velocity model is shown by a solid blue line. The dashed red line shows a $q(t)$ for an unconstrained oscillator with the same amplitude $q_0$.

The motion of the constrained oscillator is periodic. For $q_0 < 1$, the period is constant and equal to $2\pi$. For $q_0 \geq 1$, the period is

$$T = 4\left[\arcsin(1/q_0) + \sqrt{q_0^2 - 1}\right].\tag{33}$$

Let us note that the period $T$ is always larger than $2\pi$. For $q_0 \gg 1$, $T \simeq 4q_0$.

It is instructive to represent a motion of an oscillator in the limiting velocity model using a 3D phase space with coordinates $(q, \dot{q}, \chi)$. Such a phase trajectory is shown in Figure 4. For the subcritical motion of the system, the control function $\chi(t)$ vanishes, and its phase trajectory lies on the $(q, \dot{q})$ plane. At this stage, the energy of the unconstrained harmonic oscillator

$$E = \frac{1}{2}(\dot{q}^2 + q^2)\tag{34}$$

is constant. The decrease of the potential energy is compensated for by the growth of the kinetic energy. After the system enters the supercritical regime, the following quantity is conserved:

$$\mathcal{E} = E + \chi\dot{q}^2.\tag{35}$$

When the velocity of the system reaches its limiting value, the kinetic energy of the system becomes "frozen", while its potential energy decreases. As a result, the energy $E$ decreases. However, at this stage the control field $\chi$ takes a non-zero value and contributes to the total energy $\mathcal{E}$ of the system. This contribution exactly compensates for the decrease of $E$. As a result, the total energy $\mathcal{E}$ during the supercritical phase remains constant. In such a description the field, $\chi$ plays the role of some "hidden variable". One can say that this hidden variable at first absorbs a part of the energy of the oscillator and later releases it. When the total absorbed energy is returned to the oscillator, the field $\chi$ vanishes and total energy $\mathcal{E}$ coincides with the standard energy of the unconstrained oscillator $E$. It happens when $\chi$ vanishes, and a transition to the subcritical regime occurs.

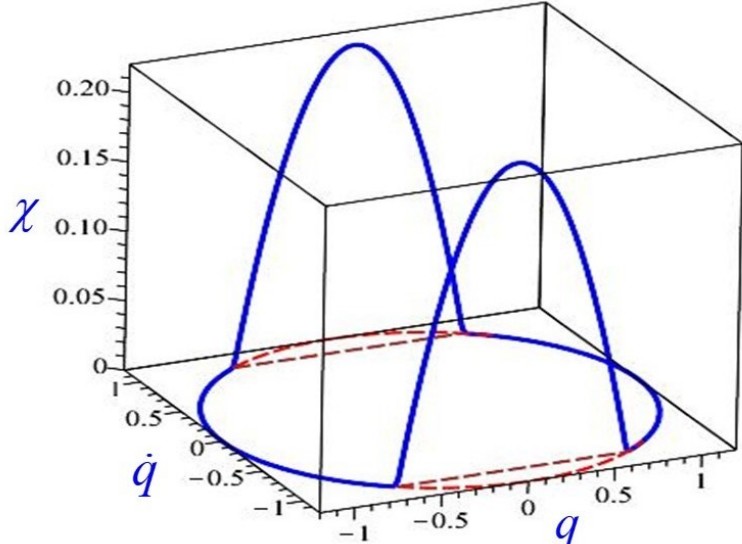

**Figure 4.** Three-dimensional phase diagram representing the motion of an oscillator in the limiting velocity model. The corresponding 3D coordinates are $(q, \dot{q}, \chi)$. This plot is constructed for $q_0 = 1.2$.

### 4.2. Modified Action Approach

Time evolution of the oscillator described in the previous subsection is periodic. The oscillator starts its motion in the subcritical regime and at point 0 enters the supercritical regime. It cannot continue its motion along the unconstrained trajectory since this would violate the inequality constraint. The oscillator can leave the supercritical regime at point 1,

where $\chi = 0$. A natural question is why it does not continue its motion further along the constraint curve. Let us discuss this point in more detail.

Let us assume that, after point 1, the oscillator continues moving with $\dot{q} = 1$. It is easy to check that then the control function becomes negative and the energy $E$ of the oscillator becomes larger than its value $E_0$ at the point of the transition. For the subcritical branch, the energy remains constant and equal $E_0$. Therefore, at transition point 1, the oscillator should "decide" which of these two branches it "prefers". One can introduce an additional requirement that its preference always should be a choice of the branch with the smaller value of the energy $E$.

Let us show that one can achieve the same result by a simple modification of the action (20). Specifically, let us consider the following action

$$ S = \frac{1}{2} \int dt \left[ \dot{q}^2 - q^2 + \eta^2 (\dot{q}^2 - 1 + \zeta^2) \right] . \tag{36} $$

It is identical to (20) with the only change $\chi = \eta^2$. The corresponding modified equations take the form

$$ (\dot{q}\eta^2)^{\cdot} = -(\ddot{q} + q) , \quad \eta(\dot{q}^2 - 1 + \zeta^2) = 0 , \quad \zeta\eta^2 = 0 . \tag{37} $$

For the subcritical regime, when $\eta = 0$, the unconstrained equations of motion are reproduced. As earlier, for the supercritical regime, when $\eta \neq 0$, one has

$$ \zeta = 0, \quad \dot{q}^2 - 1 = 0 . \tag{38} $$

Suppose, after transition point 1, the oscillator "decides" to continue its motion along the constraint. Then, one has $\ddot{q} = 0$ and at point 1

$$ (\eta^2)^{\cdot}_1 = -\frac{q_1}{\dot{q}_1} , \tag{39} $$

where index 1 indicates that a corresponding quantity is calculated at point 1. Since, at this point, $\frac{q_1}{\dot{q}_1} > 0$, the control function $\eta_1^2$ should decrease and become negative. This is impossible because $\eta^2 \geq 0$. This means that further motion along the constraint beyond point 1 is impossible and the oscillator moves from point 1 to point 2 along the subcritical trajectory, shown in Figure 2. A similar analysis, repeated for the other critical points, shows that this figure accurately represents the motion of the oscillator for the action (36).

## 5. Limiting Velocity Oscillator vs. Relativistic Oscillator

In the previous section, we described a motion of a harmonic oscillator in the model with limiting velocity. In this section, we compare this model with the case of a relativistic oscillator, where the velocity of the system is naturally restricted by a universal constant $c$ (speed of light). Using the same dimensional units as earlier, and putting $c = 1$, we write the action for such a relativistic oscillator in the form

$$ S = \int dt \, L, \quad L = - \left[ \sqrt{1 - \dot{q}^2} + \frac{1}{2} q^2 \right] . \tag{40} $$

For small velocities $|\dot{q}| \ll 1$, the Lagrangian $L$ reduces to

$$ L_{\text{nonrel}} = \frac{1}{2} \left[ \dot{q}^2 - q^2 \right] , \tag{41} $$

up to a constant term which does not affect the equation of motion. The corresponding Euler–Lagrange equation for the action $S$ is

$$ \dot{p} + q = 0, \quad p \equiv \frac{dL}{d\dot{q}} = \frac{\dot{q}}{\sqrt{1 - \dot{q}^2}} . \tag{42} $$

The conserved energy of the relativistic oscillator is

$$E \equiv p\dot{q} - L = \frac{1}{\sqrt{1-\dot{q}^2}} + \frac{1}{2}q^2 \,. \tag{43}$$

The first term in this expression, $(1-\dot{q}^2)^{-1/2}$, is nothing but the Lorentz factor $\gamma$ for a relativistic particle, which, for a unit mass, coincides with total (including the rest mass) kinetic energy of the system, while the second term, $q^2$, is the potential energy.

The motion is bounded. Let us denote the maximal value of the coordinate $q$ for a given energy $E$ by $q_0$. Then, one has

$$E = 1 + \frac{1}{2}q_0^2 \,, \tag{44}$$

and Equation (43) takes the form

$$\gamma \equiv (1-\dot{q}^2)^{-1/2} = 1 + z, \quad z = \frac{1}{2}(q_0^2 - q^2) \,. \tag{45}$$

Solving this equation for $\dot{q}$, one obtains

$$\dot{q} = \pm \frac{\sqrt{z(2+z)}}{1+z} \,. \tag{46}$$

This relationship allows one to plot a phase diagram for the motion of the relativistic oscillator in the $(q, \dot{q})$ plane. This diagram is shown in Figure 5. The velocity of the oscillator has a maximal value at $q = 0$

$$\dot{q}_0 = \pm \frac{\sqrt{z_0(2+z_0)}}{1+z_0}, \quad z_0 = \frac{1}{2}q_0^2 \,. \tag{47}$$

It is easy to check that $|\dot{q}_0| < 1$ and it becomes close to $\pm 1$ for large values of $q_0$.

By integrating the first-order differential Equation (46), one can find a position of the oscillator at time $t$, $q(t)$. To illustrate its behavior, we show in Figure 6 function $q(t)$ for $q_0 = 1.2$ (a solid blue line). For comparison, this figure also shows the function $q(t)$ for a similar non-relativistic oscillator with the same amplitude $q_0$ (a dashed red line).

The motion of the relativistic oscillator is periodic. The corresponding period is

$$T_{\text{rel}} = 2\sqrt{2} \int_0^{z_0} \frac{(1+z)\,dz}{\sqrt{z(2+z)(z_0-z)}} \,. \tag{48}$$

This integral can be calculated analytically with the following result

$$T_{\text{rel}} = \frac{4\sqrt{2}}{\sqrt{z_0+2}} \left[ (z_0+2)E\left(\sqrt{\frac{z_0}{z_0+2}}\right) - K\left(\sqrt{\frac{z_0}{z_0+2}}\right) \right] \,. \tag{49}$$

Here $E(x)$ and $K(x)$ are complete elliptic integrals of the second and first kind, correspondingly. For $q_0 \gg 1$

$$T_{\text{rel}} \simeq 4q_0 \,. \tag{50}$$

This result has a simple explanation: For a large amplitude $q_0$, the system very quickly reaches a velocity close to the speed of light, which in our units is 1, and goes from $+q_0$ to $-q_0$ and back in the course of one period.

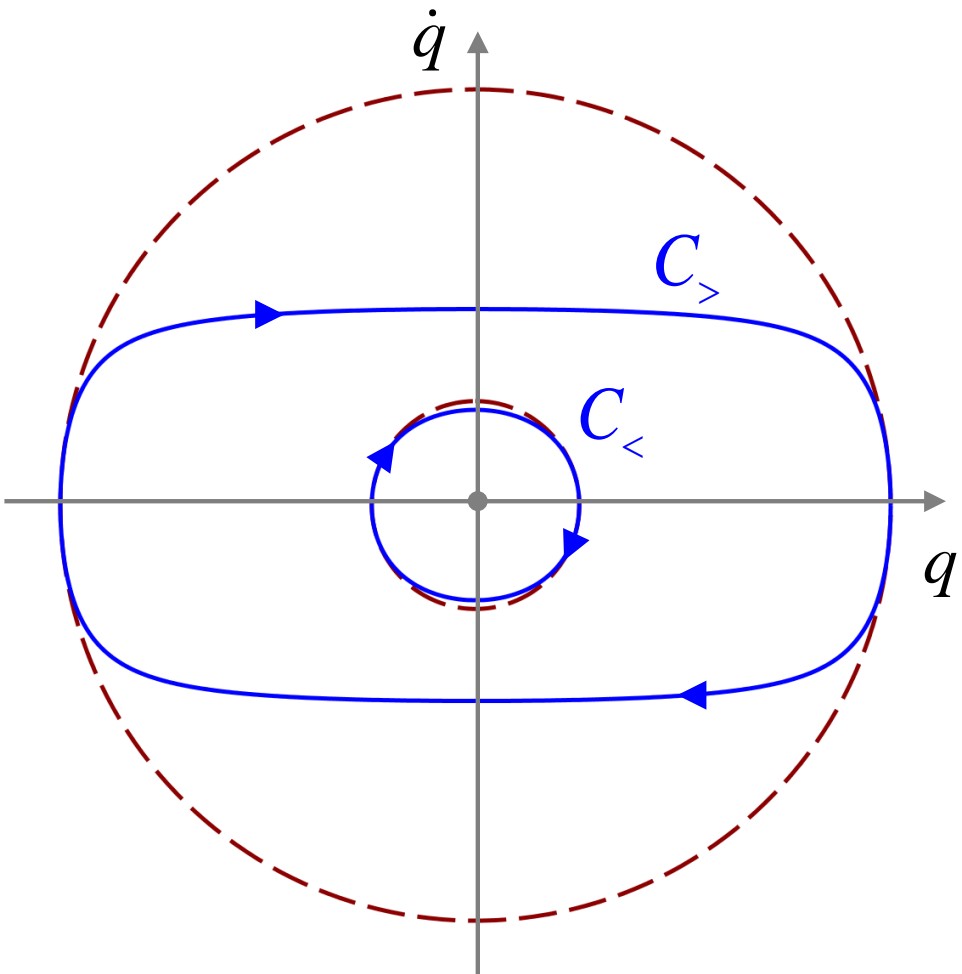

**Figure 5.** Phase diagram representing the motion of a relativistic oscillator on the $(q, \dot{q})$-plane. Two curves are shown for two values of the amplitude $q_0 = 0.5$ (line $C_<$) and $q_0 = 2.0$ (line $C_>$). For $q_0 = 0.5$ relativistic effects are small and the phase trajectory $C_<$ of the oscillator (shown by a solid blue line) is close to the cycle of radius 0.5 (shown by a dashed red line). For the amplitude $q_0$ bigger than 1 the relativistic effects are important. As a result, the phase trajectory is squashed in $\dot{q}$ direction. Such a squashed trajectory for $q_0 = 2$ is shown by a solid blue line $C_>$.

Figure 7 shows the dependence of the period $T$ of the relativistic oscillator as a function of its amplitude $q_0$ (solid red line). A dashed blue line in this plot shows the dependence of the period of the oscillator in the limiting velocity model on its amplitude $q_0$. One can see that for both cases the corresponding lines are quite similar.

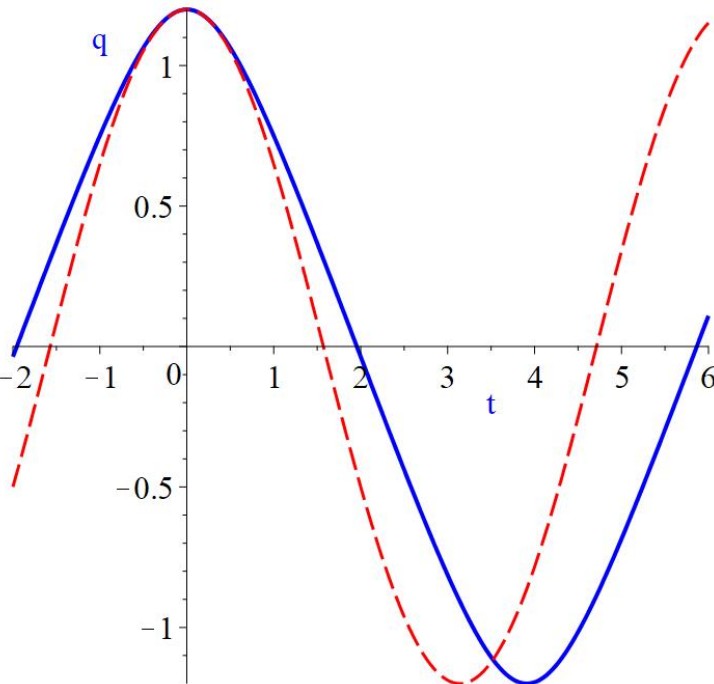

**Figure 6.** Time dependence of a position of the relativistic oscillator with amplitude $q_0 = 1.2$ is shown by a solid blue line. The dashed red line shows a position of a non-relativistic oscillator with the same amplitude.

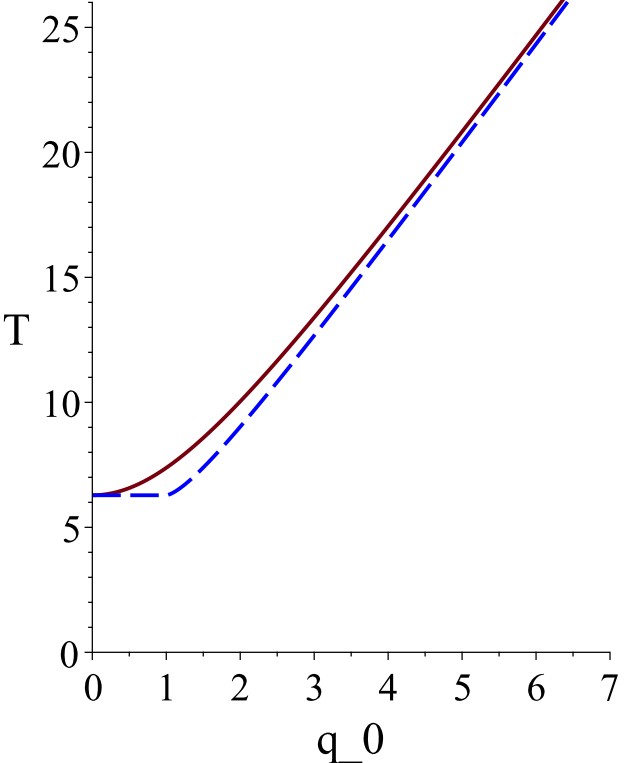

**Figure 7.** Periods of the relativistic oscillator (solid red line) and the oscillator in the limiting velocity model (dashed blue line) as functions of their amplitude $q_0$.

## 6. Inequality Constraints with Higher than First Derivatives

Till now, we have considered a case where the constraint function $\Phi$ depends only on $q$ and $\dot{q}$. Let us discuss now what happens if the constraint function contains second or

higher derivatives of $q$. Let us assume first that $\Phi$ contains only the second derivative of $q$, which enters linearly. Then, the Lagrangian $\mathcal{L}$ which enters the action (4) is

$$\mathcal{L} = L(q, \dot{q}) + \chi(\Phi + \zeta^2), \qquad \Phi = \ddot{q} - \varphi(q, \dot{q}). \tag{51}$$

Such a constraint means that the acceleration $\ddot{q}$ is restricted from above by the quantity $\varphi(q, \dot{q})$. If a subcritical solution reaches a point where $\ddot{q} = \varphi(q, \dot{q})$, it enters the supercritical regime where the following equations are valid

$$\ddot{q} = \varphi(q, \dot{q}), \qquad \frac{\delta \mathcal{L}}{\delta q} \equiv \frac{\delta L}{\delta q} - \frac{\delta(\chi\varphi)}{\delta q} - \ddot{\chi} = 0. \tag{52}$$

A trajectory $q(t)$ can be found by solving the first of these equations. The second equation defines the evolution of the control function $\chi$. It can be written in the form

$$\ddot{\chi} + \varphi_{,\dot{q}} \dot{\chi} + \frac{\delta\varphi}{\delta q} \chi = \frac{\delta L}{\delta q}. \tag{53}$$

This is a second-order linear inhomogeneous ordinary differential equation. Its solutions are uniquely determined by specifying the initial values of $\chi$ and $\dot{\chi}$ at some moment of time. In the subcritical regime, the control function $\chi$ identically vanishes. Equation (53) implies that if $\varphi_{,\dot{q}}$ and $\delta\varphi/\delta q$ are finite, then neither $\chi$ nor $\dot{\chi}$ can jump. Hence, at the transition point, one has

$$\chi = \dot{\chi} = 0. \tag{54}$$

After substitution of a solution $q(t)$ of the constraint equation into the right-hand side of (53) and using these initial conditions, one can find the time dependence of the control function in the supercritical regime.

One can check that during the supercritical phase the following quantity remains constant

$$\mathcal{E} = E - \dot{q}\dot{\chi} + [\varphi - \varphi_{,\dot{q}}\dot{q}]\chi, \qquad E = \dot{q}L_{,\dot{q}} - L. \tag{55}$$

To prove that $\dot{\mathcal{E}} = 0$ it is sufficient to use Equations (52) and (54) and the following relationship

$$\dot{E} = \frac{\delta L}{\delta q}\dot{q}. \tag{56}$$

As earlier, one can consider $\mathcal{E}$ as an "upgraded" version of the energy, which besides the energy $E$ of the original unconstrained system contains a contribution of the energy related to the control function $\chi(t)$. Motion along the constraint $\Phi = 0$ results in the change in $E$, which is compensated for by the contribution of the control parameter variable to the total energy.

Let us demonstrate that in a general case, if the solution has entered into the supercritical regime it generically cannot return to its subcritical phase. Let us suppose the opposite: at some moment of time $t_1$, the supercritical solution leaves the constraint. Since for $t > t_1$ the control function $\chi(t)$ should identically vanish, at the point of the transition one has $\chi = \dot{\chi} = 0$. If $\varphi_{,\dot{q}}$ and $\delta\varphi/\delta q$ are finite, Equation (53) implies that the functions $\dot{\chi}(t)$ and $\chi(t)$ cannot jump. However, for a general solution of (53) which vanishes at $t = t_1$, its first derivative does not necessarily vanish at this point, and the condition $\chi = \dot{\chi} = 0$ is not valid, except for the cases which form a valid boundary value problem. This property makes the case when the constraint function $\Phi$ contains second derivatives quite different from the earlier considered model with $\Phi(q, \dot{q})$.

One can expect that this is a generic property of models with inequality constraints that contain second and/or higher derivatives of $q$. In other words, if a solution enters from the subcritical regime to the supercritical one, then it is highly likely that the solution remains in the supercritical phase forever.

### 7. A Harmonic Oscillator in a Limiting Acceleration Model

To illustrate the properties of a system with an inequality constraint containing second derivatives, let us consider a simple model of a non-relativistic harmonic oscillator with a limiting acceleration. Specifically, we assume that the acceleration of the oscillator should always be smaller than some positive quantity, which we denote by $a$. We choose the corresponding constraint function $\Phi$ as follows:

$$\Phi = \ddot{q} - a \,, \tag{57}$$

and write the action in the form

$$S = \int dt \left[ \frac{1}{2}(\dot{q}^2 - q^2) + \chi(\Phi + \zeta^2) \right] . \tag{58}$$

We assume that the oscillator starts its motion at $t = 0$ with $q = 0$ and its velocity is $\dot{q} = -q_0$. Then, the equation of motion of such an unconstrained oscillator in the subcritical regime is

$$q(t) = -q_0 \sin(t) \,. \tag{59}$$

Its acceleration $\ddot{q} = q_0 \sin(t)$ grows in time. If $q_0 < a$, the oscillator acceleration always remains less than the critical one, so that its motion remains subcritical forever.

In the case where $q_0 > a$, its behavior is quite different. Denote by $t_0$ the moment of time where

$$\sin(t_0) = a/q_0 \,. \tag{60}$$

A coordinate $q$ and velocity of the oscillator at this moment are

$$q(t_0) = -a, \quad \dot{q}(t_0) = -\sqrt{q_0^2 - a^2} \,. \tag{61}$$

After $t = t_0$ the oscillator moves with a constant acceleration $a$ and one has

$$q(t) = \frac{1}{2}a(t - t_0)^2 - \sqrt{q_0^2 - a^2}\,(t - t_0) - a \,. \tag{62}$$

Equation (53) for the control function $\chi(t)$ takes the form

$$\ddot{\chi} = \ddot{q} + q \,, \tag{63}$$

where $q(t)$ is given by (62). Equation (63) can be easily integrated and one has

$$\chi(t) = \frac{1}{6}(t - t_0)^3 \left[ \frac{1}{4}a(t - t_0) - \sqrt{q_0^2 - a^2} \right] . \tag{64}$$

We used here the initial conditions $\chi(t_0) = \dot{\chi}(t_0) = 0$.

The control function becomes zero again at $t = t_1$

$$t_1 - t_0 = \frac{4}{a}\sqrt{q_0^2 - a^2} \,. \tag{65}$$

However, at this point, the velocity of the oscillator does not vanish and is

$$\dot{\chi}(t_1) = \frac{8}{3}\left[ (q_0/a)^2 - 1 \right]^{3/2} \neq 0 \,. \tag{66}$$

Therefore, the conditions $\chi(t_1) = 0$ and $\dot{\chi}(t_1) = 0$ cannot be satisfied simultaneously. This means that in the model with a limiting acceleration, the oscillator after it enters from the subcritical regime into the supercritical one remains in the supercritical regime forever.

### 8. Summary and Discussion

In this paper, we discussed the properties of dynamical systems with inequality constraints. We demonstrated that such constraints can be taken into account by a proper modification of the original action of the unconstrained theory. We first discussed a theory with Lagrangian $L(q, \dot{q})$ and inequality constraint of the form $\Phi(q, \dot{q}) \leq 0$. The dynamics of such a system is described by the extended Lagrangian $\mathcal{L} = L + \chi(\Phi + \zeta^2)$, which contains two Lagrange multipliers, $\chi(t)$ and $\zeta(t)$. The motion of a system has two regimes, sub- and supercritical ones. A transition between these regimes occurs at the points of intersection of the constant energy $E$ line and the constraint curve $\Phi$ in the phase plane $(q, \dot{q})$. For the subcritical solution, as well as at the transition point, the control parameter $\chi(t)$ vanishes. During the supercritical phase, the energy $E(q, \dot{q})$ is not conserved, while the total energy $\mathcal{E}(q, \dot{q}, \chi)$, which contains a contribution of the control function, is an integral of motion.

To illustrate the properties of dynamical systems with an inequality constraint, we considered a simple model of a harmonic oscillator with an imposed condition that its velocity cannot be larger than some fixed value. Using dimensionless units, this velocity can be chosen to be equal to 1. We showed that if the amplitude $q_0$ of the oscillator is less than 1, its periodic motion does not have a supercritical phase, while for $q_0 > 1$ it contains both phases, sub- and supercritical ones. We compared the motion of the oscillator in the limiting velocity model with the motion of a relativistic oscillator and demonstrated that their properties are quite similar. In this sense, the model with a limiting velocity constraint can be considered to be a "poor person's" version of special relativity. Certainly, this model does not pretend to substitute it, but it plays a role of a phenomenological model that captures some features of special relativity where the limiting nature of the speed of light is important.

We demonstrated that already in a simple model of the oscillator with limiting velocity, there arises an interesting question. If such an oscillator moving in the supercritical regime reaches a point where a transition to the subcritical regime becomes possible, there still exists another branch in which the supercritical motion continues. We demonstrated that a simple modification of the action allows one to escape this ambiguity. A similar problem may exist for general dynamical systems with inequality constraints. An interesting question is whether a proposed method based on the modified action can be used in such cases as well.

At the end of the paper, we discussed inequality constraints which contain higher than the first derivatives of the dynamical variables. For such systems, the differential equation for the control function is of the second or higher order in derivatives and for the transition between sub- and supercritical regimes more than one condition should be satisfied. The conditions can be satisfied either for the transition from sub- to supercritical phase or for the inverse transition. However, one can expect that a situation where there exists more than one transitions between the sub- and supercritical phases is rather special and it requires some additional conditions.

Another property of systems with higher than first derivative inequality constraints which distinguishes them from the case of the first-order constraints exists. In the theory with an inequality constraint which contains only first-order derivatives, in order to describe a motion in the supercritical regime, one needs to specify the initial value of the variables $q, \dot{q}, \chi$. However, the constraint imposes a relationship on these quantities, so that only two independent variables are present. This means that in this sense, the system still has only one degree of freedom, as it has in the subcritical regime.

For constraints with higher derivatives, the situation is quite different. For example, if the constraint contains the second derivatives $\ddot{q}$, a system in the supercritical phase is described by two second-order equations for $q$ and $\chi$. It requires four initial conditions, and in this sense, the system has two degrees of freedom, instead of one, which applies to the subcritical regime. If such a system starts its motion in the subcritical regime, then $\chi = \dot{\chi} = 0$. At the transition point to the supercritical regime, this uniquely determines the initial conditions for the control function $\chi$, and further evolution of the system depends

only on two initial value parameters $q_0$ and $\dot{q}_0$. However, if the evolution starts in the supercritical regime, it is not clear how the extra degrees of freedom should be fixed. Let us remark that a similar situation occurs in systems discussed in a recent paper [16] in a different context. Maybe this is because, in both cases, the Hamiltonian analysis is in some sense "ill-posed" (see, e.g., [17,18]).

In our discussion, we focused on systems with one degree of freedom with a single inequality constraint. It is possible to extend this analysis to more general systems that have more than one degree of freedom and several inequality constraints. One can expect that the behavior of such systems is more complicated. In particular, besides a subcritical regime, there may exist several supercritical regimes with possible transitions between them. The study of such systems is quite an interesting problem.

Let us mention that models with inequality constraints have an interesting application to the recently proposed models of gravity with limiting curvature. It is well known and widely accepted that classical General Relativity is an incomplete theory. In particular, both in cosmology and black holes, it predicts the existence of singularities under physically realistic assumptions. Famous theorems of Penrose and Hawking demonstrate that the singularities are a common property of such solutions [19–21]. For known black hole and cosmological solutions of the Einstein equations, a formation of the singularity is usually related to the infinite growth of the curvature invariants. There exist plenty of publications where different ways of modifying gravity equations were proposed to prevent curvature singularity formation. Quite a long time ago, Markov proposed a general principle of the limiting curvature [22,23] (see also [24]). According to this principle, the curvature invariants for solutions of consistent gravity equations should not infinitely grow, and their value should be restricted by some universal quantity related to a fundamental length. If such a principle is valid, then it might have quite interesting consequences, such as a possibility of a new universe formation inside black holes and bouncing cosmologies. Certainly, it would be nice to construct a modified gravity model that automatically satisfies the limiting curvature principle. However, despite many efforts, we still do not have such a theory that is consistent and satisfies everybody. In this situation, it might be useful to impose the constraints on the curvature growth by modifying the Einstein action similarly to as it was described in this paper.

Let us note that in a general case, this is not a simple problem. Even if one decides to use only the scalars, which are polynomials of the curvature, one finds that there exists quite a multitude of such independent invariants. However, for a special case of homogenous isotropic cosmologies and spherically symmetric black holes, such an approach becomes possible. This happens because in both cases, one can use the mini-superspace approach. Specifically, by using symmetries of these problems, one can present the corresponding metrics in a form that contains a small number of arbitrary functions of one variable, so that the Einstein–Hilbert action becomes a function of these metric functions. The number of the independent curvature invariants for such a mini-superspace is greatly reduced and they depend on the metric functions and their derivatives. In other words, a modified gravity with curvature constraints takes a form similar to that discussed in the present paper. Such limiting curvature models were proposed and studied in connection with the singularity problems in cosmology and in black holes in Einstein gravity in recent papers [12–15].

Let us emphasize that the basic idea of this approach is to find some robust predictions of the (still unknown) theory of gravity where curvature cannot be infinitely large. In this sense, gravity models with limiting curvature serve as some phenomenological models that can be used to study predictions of a more fundamental modified gravity theory. In some sense, they are similar to the limiting velocity model of an oscillator, which is a "poor person's" version of the complete consistent theory of special relativity. Models with inequality constraints considered in this paper are rather simple and were used only to illustrate some basic properties of such systems. It would be interesting to apply a similar approach for other physically interesting problems.

**Author Contributions:** Investigation, A.V.F. and V.P.F.; writing—original draft preparation, A.V.F. and V.P.F.; writing—review and editing, A.V.F. and V.P.F. All authors have read and agreed to the published version of the manuscript.

**Funding:** The authors are grateful to the Natural Sciences and Engineering Research Council of Canada for its financial support. V.F. also thanks the Killam Trust for its financial support.

**Acknowledgments:** The authors are grateful to the Natural Sciences and Engineering Research Council of Canada for its financial support. V.F. also thanks the Killam Trust for its financial support. The authors thank Andrei Zelnikov for his help with preparation of some of the figures.

**Conflicts of Interest:** The authors declare no conflict of interest.

## Notes

[1] Quite often, one sees a more general definition of holonomic constraints. Specifically, one calls holonomic a special class of velocity-dependent constraints, which are integrable in the sense that they can be reduced to the velocity-independent case (see e.g., [3] and references therein.).

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
