# Peer review of "Classical Mechanics with Inequality Constraints and Gravity Models with Limiting Curvature"

_universe, doi:10.3390/universe9060284_

Round 1
Reviewer 1 Report
The paper introduces a method to uplift mechanical inequalities to the level of the action. Whereas it is well-known that equalities can be lifted through the use of a Lagrange multiplier $\chi$, here, inspired by a method from optimization, a second auxiliary variable $\zeta$ is added, which appears quadratically in the action. It is shown that a strict inequality corresponds to $\chi=0$, while equality corresponds to $\zeta=0$.
The method is manifestly correct. In my opinion, it is clear and beautiful. Within the context of optimization, the authors note that the method appeared in Ref. [10] at least as early as 1996 (in fact, originally published in 1982). I am unaware of prior appearances in mechanics or physics, even after I have performed a quick literature search, yet I am not a specialist on this topic.
The paper illustrates the method through examples in mechanics and gravity. Hence, one could hardly doubt the interest and importance that this method holds.
I have a minor comment. In the first paragraph of the introduction one finds: “In the simplest case when these functions depend only on the coordinates $q$ the constraints are called holonomic”. This is correct, yet somewhat misleading, as the definition of a holonomic constraint is more general. Some authors (including Goldstein) define a constraint to be holonomic when it does not depend on velocities. Others give a more reasonable definition, where a velocity-dependent constraint is called holonomic if it can be reduced to one that is velocity-independent (which is similar to a condition of integrability). This is always the case for a constraint of the form $f(q) \dot{q}=0$, which is the case mentioned in the text.
In summary, I believe that the proposed method is correct, beautiful and useful, and I recommend publication. The authors may want to address the comment above.
Author Response
We agree with the referee and following their remark added a footnote in the Introduction clarifying the definition of the holonomic constraint. We also added a reference [3] to the book where this subject discussed in detail.
Reviewer 2 Report
The paper, ``Classical mechanics with inequality constraints and gravity models with limiting curvature,'' studies formulations of classical Lagrangian mechanics in which inequality constraints are implemented thanks to appropriate Lagrange multipliers. Examples of a non-relativistic harmonic oscillator with an implemented inequality constraint limiting either the speed or the acceleration are shown. The latter is compared to the standard relativistic harmonic oscillator, and it is shown how they yield qualitatively similar results. The paper is well written, clear, and interesting. The motivation is strong enough given the potential applicability in general relativity, with the hope of constructing a sensible theory of limited spacetime curvature. I recommend the paper for publication as is, although I would encourage the authors to shorten their title to ``Classical mechanics with inequality constraints'' since nothing is shown about ``gravity models with limiting curvature'' (beyond the level of a discussion).There are minor English mistakes here and there, but nothing that is an obstacle to the understanding of the work.
Author Response
The title of the paper was discussed with the editor and following the editor’s advice we decided to keep the present (long) version of the title.
Reviewer 3 Report
The authors discuss a possible treatment of systems with inequality constraints. The inequality (as opposed to equality) is achieved by introducing a modification of the constraint using a second power of a novel Lagrange multiplier \zeta. The resulting theory exhibits two phases: a subcritical one, where the additional constraint structure plays no role in the dynamics and a supercritical regime, which corresponds to the constraint vanishing exactly. The authors describe the conditions during which the solutions may transition between the two regimes. This novel approach is then applied on an example of a harmonic oscillator whose velocity is limited from above. They demonstrate that the resulting behavior to some degree approximates the dynamics of a relativistic oscillator, where a similar limitation to velocity is present naturally. Finally, the authors discuss the differences that arise in their approach when the constraint contains second derivatives of the variables as opposed to first derivatives. They conclude that in such cases the transition from the subcritical to supercritical regime occurs as in the previous case; however, the subsequent transition back into the subcritical regime is unlikely to occur. This is again demonstrated on an example of harmonic oscillator, with a limiting acceleration. In conclusions the authors argue that the presented approach could be very useful in the study of topics such as gravity with limiting curvature in order to avoid singularities.
The paper is overall very well written, the content is good and the extent is appropriate. However, I do have few points, which I would like to be addressed before accepting this paper for publication:
In section 3 and 4, where the change of regime is discussed, the authors claim that once the solution in the supercritical regime reaches a point where the control function vanishes, the system makes a transition from the supercritical to the subcritical regime. However, it is not clear to me why the system would be forced to do so, as it could very well continue its evolution in the supercritical regime. Furthermore, the transition from supercritical to subcritical requires a discontinuity in the evolution of the control function \chi, while by remaining in the supercritical regime its evolution is smooth. From this perspective staying in the supercritical regime seems to be the preferred option. I would appreciate if the authors could clarify this or give a comment on this.
In section 6 the evolution in the supercritical regime is determined by two second order equations of motion. To solve such equations one must provide initial conditions for both the variable q and for the control function \chi. This naively seems to imply that this regime has two degrees of freedom, while the subcritical regime only has one. When there is a transition from the subcritical to supercritical regime this extra degree of freedom is eliminated, as it was explained by the authors. However, when the system starts in the supercritical regime it is not clear how this extra freedom is fixed. A similar situation appears in systems recently discussed in arXiv:2208.05951 in a different context. It has also been argued that such systems are somewhat ill posed from the point of view of Hamiltonian analysis, see arXiv:2212.11260 and arXiv:2208.04082. I believe the paper would benefit if the authors could provide some brief commentary on this.
Some typos:
on line 33 "..describing the system motion.." should probably be "..system's.."
in the description of figure 5: "..the motion of an relativistic.." should be "..the motion of a relativistic.."
on line 272 I believe it should be velocity of the control function not the oscillator
on line 301 "derivations" should likely be "derivatives"
Author Response
We are grateful to the third referee their important comments. In connection with these comments, we made the following changes in the paper.
- We added a new subsection 4.2 where we discuss a transition from supercritical to subcritical regime. We also proposed a modification of the action for a harmonic oscillator with the limited velocity constraint which provide unambiguous description of this transition.
- We agree with the referee that there exists an asymmetry in the initial value problems in the subcritical and supercritical regimes. We also agree that this might be related to the ill-posed Hamiltonian formulation for the problem. We added a discussion of this point in Section 8. We also added 3 new references proposed by the referee.
- We corrected several typos.